# The Autoxidized Mixture of (-)-Epicatechin Contains Procyanidins and Shows Antiproliferative and Apoptotic Activity in Breast Cancer Cells

**DOI:** 10.3390/ph17020258

**Published:** 2024-02-17

**Authors:** Yazmin Osorio-Cruz, Ivonne María Olivares-Corichi, José Correa-Basurto, José Arnold González-Garrido, Fernando Pereyra-Vergara, Gildardo Rivera, José Rubén García-Sánchez

**Affiliations:** 1Laboratorio de Oncología Molecular y Estrés Oxidativo de la Escuela Superior de Medicina, Instituto Politécnico Nacional, Plan de San Luis y Díaz Mirón, s/n, Col. Casco de Santo Tomas, Ciudad de México 11340, Mexico; yaniswacky@hotmail.com (Y.O.-C.); pereyravfer@gmail.com (F.P.-V.); 2Laboratorio de Diseño y Desarrollo de Nuevos Fármacos e Innovación Biotecnológica, Escuela Superior de Medicina, Instituto Politécnico Nacional, Plan de San Luis y Díaz Mirón, s/n, Col. Casco de Santo Tomas, Ciudad de México 11340, Mexico; jcorreab@ipn.mx; 3Laboratorio de Bioquímica y Biología Molecular, Centro de Investigación de Ciencia y Tecnología Aplicada de Tabasco (CICTAT), División Académica de Ciencias Básicas, Universidad Juárez Autónoma de Tabasco, Carretera Cunduacán-Jalpa KM. 1 Colonia la Esmeralda, Villahermosa 86690, Mexico; arnold.gonzalez@ujat.mx; 4Laboratorio de Biotecnología Farmacéutica, Centro de Biotecnología Genómica, Instituto Politécnico Nacional, Reynosa 88710, Mexico; gildardors@hotmail.com

**Keywords:** procyanidins, breast cancer, apoptosis, antiproliferative activity, (-)-epicatechin

## Abstract

For this study, procyanidins generated through the autoxidation of (-)-epicatechin (Flavan-3-ol) under mildly acidic conditions (pH = 6.0) were characterized with ultra high-performance liquid chromatography (UHPLC) coupled with tandem mass spectrometry (MS/MS). Two procyanidins (types A and B) and a mix of oligomers were generated through the autoxidation of (-)-epicatechin. The antiproliferative activity of this mixture of procyanidins on MDA-MB-231, MDA-MB-436, and MCF-7 breast cancer cells was evaluated. The results indicate that the procyanidin mixture inhibited the proliferation of breast cancer cells, where the activity of the procyanidin mixture was stronger than that of (-)-epicatechin. Moreover, the mechanism underlying the antiproliferative activity of procyanidins was investigated. The resulting data demonstrate that the procyanidins induced apoptotic cell death in a manner selective to cancerous cells. In particular, they caused the activation of intrinsic and extrinsic apoptotic pathways in the breast cancer cells. The findings obtained in this study demonstrate that the generation of procyanidins in vitro by the autoxidation of (-)-epicatechin has potential for the development of anti-breast cancer agents.

## 1. Introduction

Polyphenols are the most common phytochemicals in nature and bioactive components in our diet [1]. They are highly present in fruits, vegetables, some cereals, legumes, cocoa, and beverages (wine, tea, and coffee). Polyphenols are classified according to their chemical structure in phenolic acids, stilbenes, lignans, and flavonoids [2].

Flavonoids are bioactive compounds that have attracted attention due to their potential therapeutic effect in Alzheimer’s disease [3] and cerebrovascular alterations [4], in addition to their use in medicine for antiangiogenic [5], antitumor [6], neuroprotective, antiproliferative, and anticancer functions [7].

Another class of polyphenols is generally referred to as proanthocyanidins, also known as condensed tannins or oligomeric flavonoids. It has been evidenced that these oligomeric flavonoids have a stronger biological activity than monomers [8,9], which correlates positively with their degree of polymerization [10,11,12]. Proanthocyanidins are effective against cancer, inflammation, cardiovascular disease, type 2 diabetes, and autoimmune diseases because of their powerful antioxidant activities, as well as their targeted protein-binding and cell signaling pathway regulating abilities [13,14,15].

Chemically, the proanthocyanidins differ from each other in (1) the number and position of hydroxyl groups linked to the B-ring (aromatic ring) (see Figure 1A), (2) the stereochemistry and position of hydroxyl groups on the C-ring, and (3) the linkage types among the different units. According to the above, three main proanthocyanidin types have been identified: propelagordins, procyanidins, and prodelphinidins, having one, two, and three hydroxyl groups in the B-ring, respectively. Specifically, procyanidins are relevant in the human diet, being present in different foodstuffs, such as apples, grapes, tea, berries, fruit juices, and cocoa-based products [9,16,17].

In cocoa and cocoa-derived products, the flavanol (-)-epicatechin constitutes up to 35% of the total number of polyphenols [18,19], and the procyanidins comprising catechin and (-)-epicatechin constitute 58–65% of the total number of polyphenols. Indeed, these procyanidins can be found in the oligomeric form with a degree of polymerization equal to 10; however, the predominant procyanidins are dimers to hexamers [20]. Furthermore, reports have indicated that the dimeric form of these procyanidins has biological activity, including anticancer activity [21,22,23].

The process by which polymerization occurs in plants is ambiguous; indeed, it is not clear whether is an enzymatic process or occurs spontaneously [24]. It is interesting that all hypotheses on the generation of procyanidins include the existence of a flavonoid carbocation [25]; however, this flavonoid carbocation is a labile and unstable molecule, which has never have been isolated from plants. Analyses of chemical synthesis in vitro have indicated that flavan-3-ol intermediates that undergo acid-catalyzed degradation could produce the flavan-3-ol carbocations [26,27].

The polymerization reactions of catechin have been investigated with different methods, such as enzymatic and chemical oxidation or autooxidation, which generate a group of oligomers with different chemical structures than the natural species [28,29].

These oligomers are generated through the oxidation of the monomeric units of (+)-catechin and (-)-epicatechin, which are linked by inter-flavonoid carbon–carbon bonds between C-4 and C-8 or C-4 and C-6. Interestingly, it has been described that the difference between natural dimers and dimers generated by oxidation is in the bond position (C6 to C8 or C6 to C6) [30,31]. Furthermore, these procyanidins are considered to be a group of unnatural oligomers, which have the same molecular ion mass spectra (MS1) but different structures when compared to natural isomeric procyanidins.

Significant developments have been made regarding the structural information of natural procyanidins, through of ultra-high-pressure liquid chromatography (UHPLC) coupled with mass spectrometry (MS) and/or nuclear magnetic resonance (NMR) [32,33,34]. However, little is known about the synthesis, structure, and biological activity of synthesized unnatural procyanidins. In this context, research on synthesized procyanidins has become important, because, in many cases, they have various biological activities including anticancer activities [35]. In addition, we considered that the procyanidins obtained from natural sources required a process of purification, which, in many cases, is difficult, the synthesis of procyanidins of the mixer generation with a stronger activity than natural procyanidins has become an option in the generation of molecules with therapeutic potential. The aim of this study was to synthesize unnatural procyanidins to establish their structure, activity, and possible mechanism of action in breast cancer cells. Oligomers of (-)-epicatechin were synthesized using monomeric units of (-)-epicatechin that were autoxidized in aqueous/methanol model solutions under mildly acidic conditions (pH 6.0). A chemical characterization of the oxidation products was performed through UHPLC-TOF/MS/MS (LC-MS) analysis. Furthermore, the antiproliferative activity and its possible mechanism of action were established in different breast cancer cells.

## 2. Results and Discussion

### 2.1. Autoxidation of (-)-Epicatechin

The generation of the products of the polymerization oxidation (PPO) from (-)-epicatechin was carried out through autoxidation (Figure 1A). Initially, the autoxidation of this flavonoid was monitored with thin layer chromatography (TLC) throughout the reaction, at 6, 12, 24, 48, and 72 h. Changes in TLC product mobility were detected at 72 h, and the presence of a band close to the origin of the application was observed (Figure 1B). The autoxidation of (-)-epicatechin under this condition (mildly acidic conditions pH = 6.0) was analyzed with LC-MS.

#### Identification of the Molecules Present in the PPO Mix

The chromatographic profile of the autoxidation of (-)-epicatechin yielded eight products (Figure 2A). For each product, positive MS and MS2 spectra were obtained, which were used to perform an identification according to the literature. Figure 2B shows the *m*/*z* of the compounds detected and their fragmentation. Peaks 1 and 2 were unknown compounds, the *m*/*z* of which is shown in Figure 2B. Peaks 3 and 4 showed the same (M + H)+ ion and the MS/MS spectra of these ions presented the same ionic products (Figure 2B). These data indicated the presence of (-)-epicatechin (residual substrate; Figure 2A, peak 4) and catechin (converted from (-)-epicatechin; Figure 2B, peak 3). It is important to mention that both compounds were distinguished via their different retention times [36,37]. On the other hand, the peak 5 [M + H]+ ion at *m*/*z* 579 and the MS/MS spectrum suggested a procyanidin dimer with a β-type linkage [31,38]. Peak 6 (Figure 2A) was a dimer (Procyanidin A), according to the MS/MS spectrum (Figure 2B) [35]. The other two unknown products (peaks 7 and 8) were also observed, and their *m*/*z* is shown in Figure 2B. It is important to note that the dimers generated via the oxidation of (-)-epicatechin could differ in the position of interflavan linkage (IFL), compared to that observed in natural dimers; still, they presented the typical [M + H]+ molecular ions of B-type and A-type procyanidins (Figure 2B) [30,31]. As the autoxidation of (-)-epicatechin for 72 h generated a band with different TLC mobility (Figure 1B), this band was purified and analyzed with electrospray ionization (ESI) (Figure 2C). The analysis of this band showed the presence of several peaks, which were in low concentration. For this reason, an analysis of the polymeric distribution was performed in the region of 200–2000 *m*/*z* (Figure 2C). This distribution showed a majority peak with an *m*/*z* of 411.09 (Figure 2C, peak 9). On the other hand, the peak with *m*/*z* 865.1975 suggested the generation of the trimer of (-)-epicatechin (Figure 2C, peak 11). Other peaks (Figure 2C, peaks 9, 10, 12–15) may indicate the presence of (-)-epicatechin-derived products of different molecular weights, but their structures could not be identified.

Figure 3 shows the chemical structure of the monomers, the procyanidins identified, and the possible oligomers generated. With these data, it is worth noting that the autoxidation of (-)-epicatechin under mildly acidic condition allows for the formation of a complex mixture of PPO or procyanidins.

Although MS data alone were used to establish the generation of procyanidins, we also investigated whether this mixture of PAs generated in vitro could have similar biological activities as natural procyanidins.

### 2.2. PPO Mixture Inhibited Cell Proliferation of Breast Cancer Cells

Several studies have reported the anticancer activity of (-)-epicatechin and its procyanidins [39,40]. Our interest was in establishing this activity for the generated mixture of PPO. As shown in Figure 3, MDA-MB-231, MDA-MB-436, and MCF-7 breast cancer cells, as well as MCF-10A (normal cells), were grown in the presence of different concentrations of (-)-epicatechin, the generated PPO (0–350 µM), or autoxidation buffer as a control for 72 h (same volume used in each treatment). We found that (-)-epicatechin suppressed MDA-MB-231 cell proliferation in a concentration-dependent manner (Figure 4A). Indeed, the highest concentration of epicatechin (350 µM) decreased cell viability to 34.5%. However, with the generated PPO mix, the greatest inhibition (32% viability) was observed with the lowest concentration used (50 µM), and this effect was maintained at higher concentrations (Figure 4A). In addition, no effect was observed with the autoxidation buffer. Interestingly, in the triple-negative breast cancer MDA-MB-436 cells, (-)-epicatechin decreased cell viability to 80% in all concentrations used, while PPO treatment decreased cell viability to 33%, with a maximum effect at 200 µM (Figure 4B). On the other hand, in the estrogen-positive MCF-7 cells, (-)-epicatechin inhibited cell proliferation in a concentration-dependent manner, with the highest growth inhibition (29% cell viability) at the highest concentration of (-)-epicatechin used (350 µM). Nevertheless, PA treatment showed the highest effect (22% cell viability) at 200 µM (Figure 4C). Of note, when the cytotoxic activity of (-)-epicatechin and the PPO mix were analyzed at the same concentrations in the non-cancerous breast cell line MCF-10A, no cytotoxic activity was observed (Figure 4D). These data indicate that (-)-epicatechin and the mixture of generated procyanidins exhibited selective cytotoxic activity against cancerous cells (Figure 4D).

### 2.3. Morphology Changes and DNA Fragmentation Induced by PPO Mix in MDA-MB-231 Breast Cancer Cells

As MDA-MB-231 cells showed a higher sensitivity to PPO, they were selected for subsequent determinations. First, to determine whether the antiproliferative activities observed with (-)-epicatechin and the PPO mix were related to apoptosis, we characterized their effects on cell morphology and DNA fragmentation, which are features of the apoptotic process and do not indicate necrotic activity [41]. Figure 5 shows the morphological changes that were generated after 72 h of treatment with the autoxidation buffer (MES-NaOH), (-)-epicatechin (350 µM), and the PPO mixture (350 µM). The cells that were treated with autoxidation buffer showed a morphology that was characteristic of MDA-MB-231 cells (Figure 5A). However, a spherical morphology was observed in cells treated with (-)-epicatechin (Figure 5B); additionally, PPO treatment resulted in membrane blebs (Figure 4C, arrows). On the other hand, the presence of DNA and its analysis under these conditions demonstrated the presence of DNA fragmentation (Figure 5D)—an event that is characteristic of apoptosis. Importantly, similar results were observed in the other cell lines studied.

### 2.4. The Expression Pattern of Apoptosis-Related Proteins Activated by PPO Mixture

As the generated PAs were found to suppress cell growth and induce apoptosis, we investigated the molecular basis of the apoptotic effect in MDA-MB-231 breast cancer cells. A human apoptosis array was performed to establish a profile of the apoptosis proteins that were generated via treatment with the PPO mix. Figure 5A shows the profile in the presence of the autoxidation buffer, and Figure 6B shows the profile after treatment with the PPO mixture.

We observed that the PPO mixture generated changes in the expression of the genes involved in the extrinsic pathway of the activation of apoptosis. For example, an increase was observed in the expression of SMAC/Diablo (Figure 6B, lines D15, D16), which is an antagonist of the inhibitor of apoptosis proteins that facilitates the release of cytochrome c (Figure 6B, lines B23, B24) from the mitochondria into the cytoplasm, therefore inducing the formation of the apoptosome and consequently activating the apoptotic process [42,43]. Furthermore, PA treatment also generated an increase in the expression of the heat shock proteins HSP 27, 60, and 70, (Figure 6B, lines C15–C20) suggesting the effect of lethal toxicity in cancerous cells [44]. On the other hand, we observed that PA treatment increased the expression of death receptor 4 (TRAIL R1/DR4) and death receptor 5 (TRAIL R2/DR5) (Figure 6B, lines C1–C4), which exist on the cell surface and trigger the extrinsic and intrinsic apoptotic pathways [45]. Interestingly, these receptors selectively induce apoptosis in malignant cells but not in normal cells. Finally, the phosphorylation of p53 in s15, s46, and s392 was observed (Figure 6B, spots in the squares), which strongly suggests that a key mechanism is responsible for activating the p53 tumor suppressor in response to the cellular stress generated by PAs [46]. These findings suggest that, in MDA-MB-231, a triple-negative breast cancer cell, PAs induced the apoptotic effect through both the extrinsic and intrinsic apoptotic pathways.

As PPO treatment generated an apoptotic profile in breast cancer cells, and no cytotoxic activity was observed in the MCF-10A non-cancerous cells (Figure 4D), we analyzed whether this selectivity was related to the induction of this apoptotic profile. Figure 5C,D show the apoptotic profiles generated on MCF-10A cells. No significant changes were observed in the apoptotic profile, indicating that this selectivity was generated by the capacity of the PPO mix to activate apoptosis in breast cancer cells.

## 3. Materials and Methods

### 3.1. Materials

Acetone, toluene, formic acid, (-)-epicatechin, methanol, 2-(N-morpholino ethanesulfonic acid (MES), hydrocortisone, 3-(4,5-dimethylthiazol-2-yl]-2,5-diphenyl tetrazolium bromide (MTT), and dimethyl sulfoxide (DMSO) were purchased from Sigma-Aldrich, St. Louis, MO, USA.

### 3.2. Autoxidation of (-)-Epicatechin

The autoxidation of (-)-epicatechin was performed according to the method of He et al., with some modifications [30]. Briefly, a methanolic solution of 10 mM (-)-epicatechin was generated. Additionally, 2-(N-morpholino)ethanesulfonic acid (MES)-NaOH buffer (100 mM, pH 6.0) was prepared for use as a reaction buffer. For the autoxidation of (-)-epicatechin, the methanol solution (1 mL, 10 mM) was mixed with the MES-NaOH buffer (9 mL, 100 mM) to prepare the reaction system at pH 6.0. The mixed reaction (10 mL) was incubated in the dark at 45 °C for 6, 12, 24, 48, and 72 h. The autoxidation of (-)-epicatechin was monitored with TLC and analyzed with LC-MS.

Thin-layer chromatography was performed on pre-coated silica gel 20 cm × 20 cm plates (Merck Millipore, Burlington, MA, USA). The plates were cut to 4 cm × 10 cm. Standard solutions of (-)-epicatechin and the reaction mixture were applied as 5 mm bands. The application volume for (-)-epicatechin and the reaction mixture was 0.5 µL. The developing solvents consisted of toluene, acetone, and formic acid (3:7.5:1 v:v:v). The developed plates were dried at room temperature and then immersed for 1 s in vanillin (1%) in sulfuric acid in methanol (0.3 N). Drying at room temperature formed colored bands, which are denoted as products of polymerization oxidation (PPO) in this study.

### 3.3. UHPLC-TOF/MS Analysis of Products of Polymerization Oxidation

The PPO obtained after 72 h of epicatechin autoxidation was analyzed using an Agilent 1290 UHPLC coupled with an Agilent 6545 Q-TOF/MS (Agilent Technologies, Santa Clara, CA, USA). A 15-microliter sample solution was separated on a Zorbax Eclipse Plus C18 column (1.8 µM, 2.1 X 50 mm) and maintained at 35 °C with (A) 0.1% formic acid and (B) acetonitrile with 0.1% formic acid as the mobile phase at a flow rate of 0.4 mL/min. For molecular identification there was used ESI in positive mode. The optimized conditions were capillary voltage 4 kV, fragmentor voltage 120, and the mass range was set from 100 to 1700 *m*/*z*. In addition, the activation energy for the MS/MS experiment was set to 1.0 V.

The band that was generated after 72 h of epicatechin autoxidation was cut from the plates (Figure 1B), re-suspended in methanol, and incubated overnight in a rocking platform shaker. The suspension was filtered and concentrated via evaporation. The 15-microliter sample solution was then analyzed under the conditions mentioned above.

MS/MS experiments were performed to recognize fragmentation patterns by acquisition mode untargeted MS2 and fixed collision energy of 20 eV. Chromatographic data were processed using MassHunter Qualitative Analysis Software (Version B.08.00). Compounds were extracted from the raw data using the Molecular Feature Extraction (MFE) algorithm, and compound identification was performed using the METLIN Database (www.metlin.scripps.edu, accessed 17 August 2023) and Molecular Formula Generation software, Version 8 (Agilent Technologies).

### 3.4. Cell Cultures

The cancer cell lines used in this study were obtained from the American Type Culture Collection (ATCC; Manassas, VA, USA). They consisted of MDA-MB-231, MDA-MB-436 (ER-, PR- and HER2-), and MCF-7 (ER+, PR+ and HER2+) breast cancer cells, and the MCF-10A cell line (nontumorigenic epithelial cell line) was included to represent normal cells.

MDA-MB-231 and MCF-7 were grown in high-glucose DMEM supplemented with 5% fetal bovine serum (FBS, Biowest, Miami, FL, USA), while MDA-MB-436 cells were grown in DMEM/F12 supplemented with 10% FBS (Biowest, Miami, FL, USA).

MCF-10A cells were cultured in DMEM/F-12 supplemented with 5% horse serum (Biowest, Miami, FL, USA), 20 ng/mL epidermal growth factor (Upstate Biotechnology Incorporated, Lake Placid, NY, USA), 10 µg/mL insulin (Biofluids, Rockville, MD, USA), and 500 ng/mL hydrocortisone.

All media (Biowest, Miami, FL, USA) were also supplemented with 2 mM glutamine, 100 U/mL penicillin, and 100 mg/mL streptomycin. The cells were maintained in a humidified incubator at 37 °C with 5% CO_2_.

### 3.5. Treatment of Breast Cancer Cells with (-)-Epicatechin and PPO

When the effects of (-)-epicatechin and PPO were examined, the cells were grown in their respective phenol-red free media. Under these conditions, any possible effect of phenol-red on (-)-epicatechin or the PPO mix was avoided. The concentration of PPO (procyanidins generated) was calculated based on the concentration of (-)-epicatechin used in the autoxidation reaction (10 mM).

### 3.6. Cytotoxicity Assays

MTT [3-(4,5-dimethylthiazol-2-yl]-2,5-diphenyl tetrazolium bromide] assays were performed to measure the growth of all cell lines (MDA-MB-231, MDA-MB-436, MCF-7, and MCF-10A) in the presence of (-)-epicatechin, the reaction buffer [MES-NaOH (100 mM, pH 6.0)], and the PPO mixture [47].

The cell lines (1 × 10^4^ cells per well) were seeded in 96-well plates and allowed to attach overnight in a CO_2_ incubator. The cells were treated with (-)-epicatechin and PPO (0–350 µM) for 72 h. MTT solution (1 mg/mL) was added to each well after aspirating the medium with the treatments, and the plates were incubated for 2 h at 37 °C and 5% CO_2_. After the medium had been discarded, the formazan crystals were dissolved in 150 µL of dimethyl sulfoxide (DMSO; Sigma-Aldrich-Mexico, Toluca, State of Mexico). The resulting solution was measured using spectrophotometry with a microplate reader (iMark™ Microplate Absorbance Reader, Bio-Rad Laboratories, Hercules, CA, USA) at a wavelength of 550 nm. The quantity of formazan produced was directly proportional to the number of living cells. Three different experiments were performed, each data point was obtained in sextuplicate. The results are reported as the percentage of cells viability ± standard deviation (SD) in relation to the negative control, the viability of which was designated 100%.

### 3.7. Assessment of Apoptosis Using DNA Fragmentation

The presence of inter-nucleosomal DNA cleavage in breast cancer cells (1 × 10^6^ cells/60 mm dish) grown in the presence of (-)-epicatechin (350 µM) and PPO mixture was investigated using DNA gel electrophoresis. After treatment, cell lines (MDA-MB-231, MDA-MB-436, MCF-7, and MCF-10A) were harvested with PBS-EDTA (5 mM) from the culture dishes and washed twice in PBS. They were lysed in a solution containing 100 mM NaCl, 10 mM Tris-HCl, 25 mM EDTA at pH 8, and 0.5% SDS, and incubated for 2 h at 37 °C with 0.1 mg/mL proteinase K and 200 mg/mL RNase A. Finally, DNA was extracted using phenol/chloroform [48]. The DNA (5 μg/lane) was electrophoresed on 2% agarose gels and visualized with ethidium bromide staining.

### 3.8. Human Apoptosis Antibody Array

The changes in the expression pattern of human apoptosis-related proteins that were generated by the PPO mixture were investigated using a human apoptosis array kit (R&D Systems, OR, USA), in accordance with the manufacturer’s instructions. MDA-MB-231 and MCF-10A (1 × 10^7^ cells) cells treated with oxidation buffer and PPO were solubilized in lysis buffer and centrifuged at 12,000 rpm for 20 min. Protein concentrations of the resulting lysates were measured with Lowry’s method [49]. The antibody-coated array membranes were blocked in Array Buffer 1 with 1 h incubation on a rocking platform shaker. Then, the membranes were incubated with 300 µg of protein from each sample at 4 °C with gentle shaking overnight. The following day, the membranes were washed three times with 1× washing buffer on a rocking platform, then incubated with 1.5 mL of reconstituted biotinylated antibodies for 1 h on a rocking platform. Thereafter, the biotin-conjugated antibodies were removed and again washed three times with 1× washing buffer. A 1:2000 dilution of streptavidin-HRP in 1× Array Buffer was used to incubate each of the membranes for 30 min. Following incubation, the arrays were washed three times with 1× wash buffer. The membrane intensity and pixel densities were acquired using C-DiGit (LICOR, Lincoln, Nebraska, USA). Relative differences in the expression levels of each protein in untreated and treated cells were quantified by comparing the signal intensities among the array images with the Image Study Lite version 4.0 program after subtraction of background signals.

### 3.9. Statistical Analysis

All experiments were carried out in sextuplicate in three different experiments, and the results are expressed as the mean ± SD. We utilized one-way analysis of variance (ANOVA) followed by Tukey’s test. Differences were considered statistically significant when *p* ≤ 0.05.

## 4. Conclusions

The autoxidation of (-)-epicatechin under mildly acidic conditions (pH = 6) yielded a complex mixture, which presented an antiproliferative activity selective for breast cancer cells. Characterization of the mixture through LC-MS/MS identified procyanidin types A and B, as well as the presence of the trimers of (-)-epicatechin. The antiproliferative effect of the mixture in breast cancer cells was higher than the monomer of (-)-epicatechin, demonstrating that its oxidation affected the cancer activity. It is important to mention that the anticancer activity of the mixture was found to be mediated by the induction of apoptosis, and they presented selectivity for cancer cells. This study showed that oxidized (-)-epicatechin produced in vitro may be useful in cancer therapy. However, further experiments are necessary to demonstrate their safety and activity in vivo.

## Figures and Tables

**Figure 1 pharmaceuticals-17-00258-f001:**
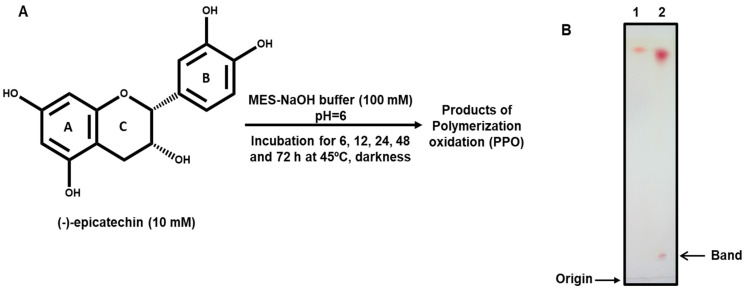
Oxidation of (-)-epicatechin: (**A**) Conditions of autoxidation of (-)-epicatechin; and (**B**) TLC pattern of (-)-epicatechin (Lane 1) and (-)-epicatechin oxidized for 72 h (Lane 2).

**Figure 2 pharmaceuticals-17-00258-f002:**
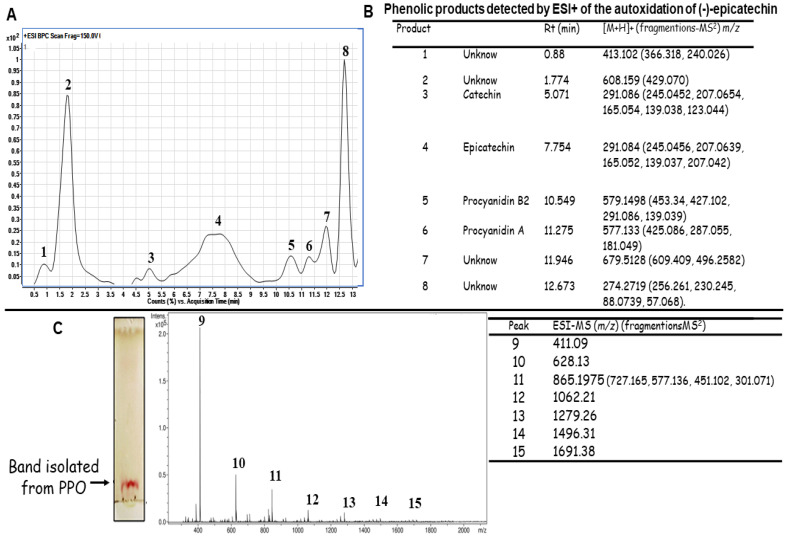
HPLC-ESI-MS^n^ spectra of the autoxidation products of (-)-epicatechin: (**A**) Base peak chromatogram of the autoxidation of (-)-epicatechin; (**B**) Products generated through the autoxidation of (-)-epicatechin; catechin [36], procyanidin B2 [31,37], procyanidin A [36] and (**C**) TLC pattern of the PPO mix and ESI-MS of the possible oligomers (peaks 9–15) obtained through the autoxidation of (-)-epicatechin.

**Figure 3 pharmaceuticals-17-00258-f003:**
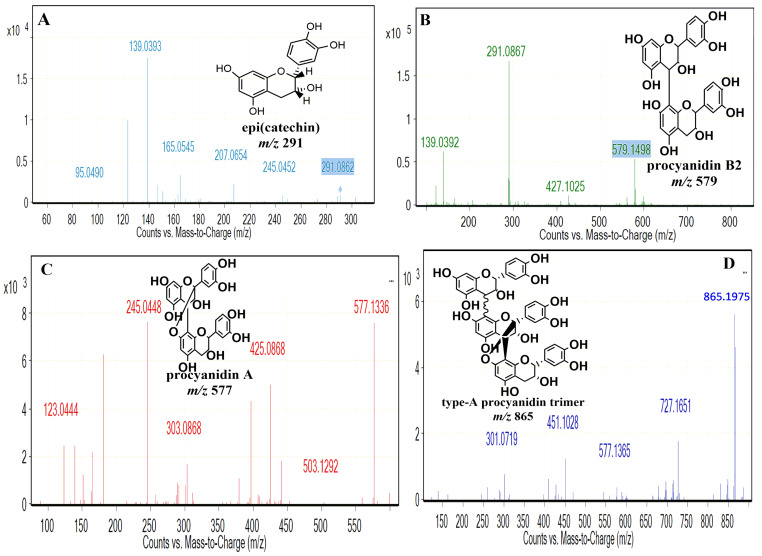
Chemical structures of the compounds present in the autoxidation of (-)-epicatechin. (**A**) Monomers (Figure 2B, peaks 3, 4); (**B**,**C**) B2 and A type procyanidin (Figure 2B, peaks 5, 6); (**D**) Type A procyanidin trimer (Figure 2C peak 11).

**Figure 4 pharmaceuticals-17-00258-f004:**
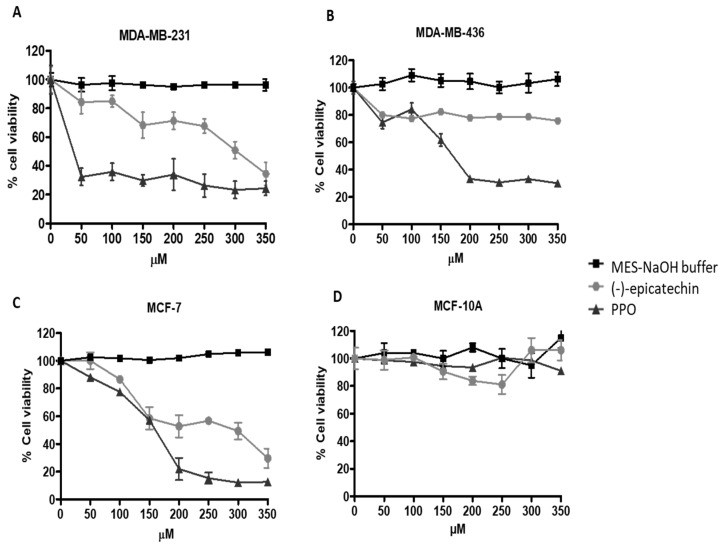
Antiproliferative activity of (-)-epicatechin and procyanidin mixture in breast cancer cells. The cells were grown in an interval of 0–350 µM: (**A**) MDA-MB-231; (**B**) MDA-MB-436; (**C**) MCF-7; and (**D**) non-cancerous MCF-10A cells. Each experiment was conducted three times, and each data point was performed in sextuplicate. Data are presented as the mean ± SD and were analyzed using ANOVA followed by Tukey’s test (*p* ≤ 0.05).

**Figure 5 pharmaceuticals-17-00258-f005:**
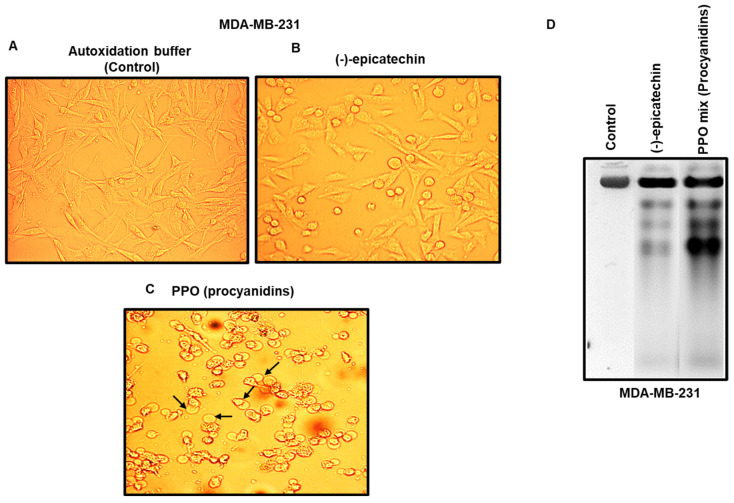
Effects of (-)-epicatechin and PPO mix in MDA-MB-231 breast cancer cells. Morphology of MDA-MB-231 breast cancer cells in the presence of: (**A**) Autoxidation buffer; (**B**) (-)-epicatechin; and (**C**) PPO mix; the arrows point out membrane blebs; and (**D**) induction of DNA fragmentation by (-)-epicatechin and PPO mix.

**Figure 6 pharmaceuticals-17-00258-f006:**
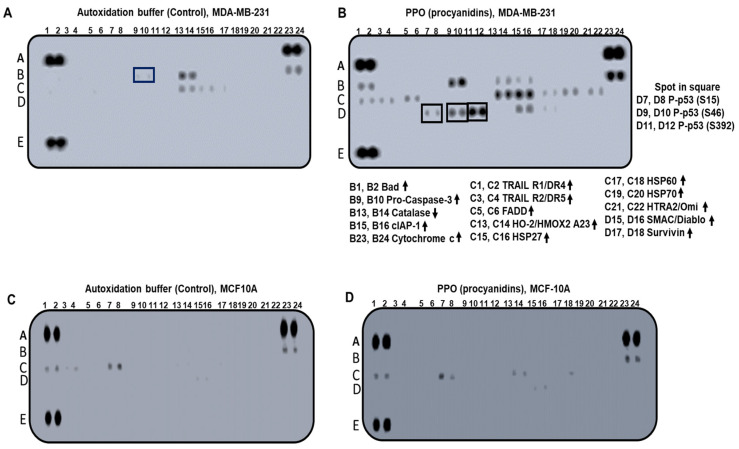
Human apoptosis array analysis in response to (-)-epicatechin and PPO treatment. Whole-cell lysates were prepared from MDA-MB-231 and MCF-10A cell lines and were left either untreated or exposed to MES-NaOH buffer and hybridized to a human apoptosis array kit (**A** and **C**, respectively). The MDA-MB-231 (**B**) and MCF-10A (**D**) cell lines were treated with PPO (proanthocyanidins) for 72 h. Each protein was spotted in duplicate. The pair of dots in each corner were positive controls. Each pair of protein dots is indicated with its change in expression. At the same time, no obvious change was observed in MCF-10A cells (**C**,**D**). Up and down arrows (Increase or decrease in the expression, respectively), the A-E letters on the left side of each membrane are coordinates for the protein identification.

## Data Availability

Data is contained within the article.

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
