# Peer review of "The Autoxidized Mixture of (-)-Epicatechin Contains Procyanidins and Shows Antiproliferative and Apoptotic Activity in Breast Cancer Cells"

_pharmaceuticals, 2024, doi:10.3390/ph17020258_

Round 1

Reviewer 1 Report

Comments and Suggestions for Authors

Dear Author,

1. The author didn't state the research question in the introduction: why has autoxidation been done? is it to increase the efficacy, or else must be clearly stated in the background about the objective this study?

2. Provide a few kinds of literature about the autoxidation of flavonoids or procyanidins and how they are better than the original compound.

3. Is there any reference for the autoxidation of f (-)-epicatechin, in methodology, if yes include the standard method as a reference

4. Increase the clarity of Figure 2C) 110 TLC pattern of the PPO mix, it looks blurred

5. Is there any study to confirm the stability of polymerized oxidation products?

6. Are there any results of cytotoxicity presented in IC50?

Thanks

Author Response

Dear Referees, thank you very much for your observation and the spend time in our manuscript, we appreciated your interest in improved our manuscript.

Reviewer 1

  1. The author didn't state the research question in the introduction: why has autoxidation been done?

R= In the lines 57-63, we briefly mention the controversy that exists about of the polymerization, and how chemical synthesis in vitro contributes at this knowledge.

is it to increase the efficacy, or else must be clearly stated in the background about the objective this study?

R= In the lines 38-41, we included evidence on degree polymerization and biological activity. At the end of the introduction (lines 77-84); it was described the objective of the study.

Provide a few kinds of literature about the autoxidation of flavonoids or procyanidins and how they are better than the original compound.

R= Evidence on the autoxidation (references 15-20), degree polymerization and activity were included (references 1-5, 12-14).

  1. Is there any reference for the autoxidation of (-)-epicatechin, in methodology, if yes include the standard method as a reference.

R= The references were included in the methodology (Reference 21).

  1. Increase the clarity of Figure 2C) 110 TLC pattern of the PPO mix, it looks blurred

R= The figure was improved

  1. Is there any study to confirm the stability of polymerized oxidation products?

R= Yes, the manuscript of Zhu QY reported the stability of the cocoa monomers, (−)-epicatechin, (+)-catechin, and dimers, at different pH values (gastric and intestinal juice). They showed that the dimers were less stable than the monomers at both acidic and alkaline pH. Interestingly, the incubation of Dimer B2 and Dimer B5 in simulated gastric juice (pH 1.8) or acidic pH resulted in degradation to epicatechin and isomerization to Dimer B5 and Dimer B2, respectively. The incubation in simulated intestinal juice or at alkaline pH, monomers and dimers were degraded almost completely within several hours

(Zhu QY, et al., J Agric Food Chem. 2002 Mar 13;50(6):1700-5. doi: 10.1021/jf011228o).

The evidence of Toro-Uribe shows that all the procyanidins underwent depolymerization into small oligomers and (epi)catechin monomers. Indeed, they generated liposome formulations, which presented higher bioaccessibility and antioxidant activity in comparison to non-encapsulated form. Similar results were obtained with procyanidins from cocoa extract-loaded liposomes. They concluded that liposomes are efficient carriers to stabilize and transport procyanidins (Toro-Uribe S, et al., J Agric Food Chem. 2019. 20;67(7):1990-2003. doi: 10.1021/acs.jafc.9b00351).

  1. Are there any results of cytotoxicity presented in IC50?

R= From our knowledge, few studies have been performed using this method of autoxidation, however, do not include the studies on the functionality of the products generated. Indeed, our manuscript is the first that include this kind of analysis. 

He, F., et al.(2009). Identification of autoxidation oligomers of flavan-3-ols in model solutions by HPLC-MS/MS. Journal of mass spectrometry : JMS, 44(5), 633–640. https://doi.org/10.1002/jms.1536

Reviewer 2 Report

Comments and Suggestions for Authors

The manuscript "The autoxidation of (-)-epicatechin generates a mix of procyanidins with antiproliferative and apoptotic activity in breast cancer cells", devoted to the analysis of cytotoxic activity of procyanidins to malignant breast cancer cells looks quite interesting in this field. The text is concise and logically laid out. The results are convincingly and clearly presented. I received favourable impressions when I reviewed this paper.  Illustrations are carefully prepared, adequately reflecting the findings of the study.

As a recommendation, I can only suggest expanding the Introduction section, which now looks too short.

It is also unclear how the results were statistically processed. This subsection should be included to the Materials and Methods section.

Author Response

Dear Referees, thank you very much for your observation and the spend time in our manuscript, we appreciated your interest in improved our manuscript.

Reviewer 2

The manuscript "The autoxidation of (-)-epicatechin generates a mix of procyanidins with antiproliferative and apoptotic activity in breast cancer cells", devoted to the analysis of cytotoxic activity of procyanidins to malignant breast cancer cells looks quite interesting in this field. The text is concise and logically laid out. The results are convincingly and clearly presented. I received favourable impressions when I reviewed this paper.  Illustrations are carefully prepared, adequately reflecting the findings of the study.

As a recommendation, I can only suggest expanding the Introduction section, which now looks too short.

R= The introduction was expanded as was recommended.

It is also unclear how the results were statistically processed. This subsection should be included to the Materials and Methods section.

R= The section of statistical analysis was included in material and methods (section 3.9), also in the figure 3 (in the figure legend) was mentioned the statistical analysis performed.

Reviewer 3 Report

Comments and Suggestions for Authors

The manuscript oxidized epicatechin, characterized the product and evaluated the anti-cancer activity.

The characterization of oxidized product is not sufficient. The area of Procyanidin B and A in the Figure 2A show that they are relatively minor components, and the m/z of Figure 2C does not show the existence of an oligomers (procyanidin oligomer have molecular weight of 578,  866, 1154, 1442, 1730 for B-type linkage. The m/z would be +1 for H+ adduct (or +23 for Na+) in the positive mode as described in the method section, which are different value from the observed data. Therefore, the authors' conclusion that autooxidation produced procyanidins and show anti-cancer activity cannot be drawn.

Comments on the Quality of English Language

Multiple grammatical mistakes are observed. Some contradictory sentences exists.

eg. Line 44 "Indeed, this procyanidins is found mainly in oligomeric form (degree of polymerization = 10), predominantly dimers to hexamers."
>Degree of polymerization would be the average number and it would mean that predominant compounds have polymerization degree of 10 in cocoa. This sentence also requires a reference.  

Author Response

Dear Referees, thank you very much for your observation and the spend time in our manuscript, we appreciated your interest in improved our manuscript.

Reviewer 3

Comments and Suggestions for Authors

The manuscript oxidized epicatechin, characterized the product and evaluated the anti-cancer activity.

The characterization of oxidized product is not sufficient. The area of Procyanidin B and A in the Figure 2A show that they are relatively minor components, and the m/z of Figure 2C does not show the existence of an oligomers (procyanidin oligomer have molecular weight of 578, 866, 1154, 1442, 1730 for B-type linkage. The m/z would be +1 for H+ adduct (or +23 for Na+) in the positive mode as described in the method section, which are different value from the observed data. Therefore, the authors' conclusion that autooxidation produced procyanidins and show anti-cancer activity cannot be drawn.

R= Referring to the area of Procyanidin B and A in the Figure 2A, I would like to say that in this model of oxidation the products obtained are minor, data that also have been reported by other researchers (He, F., et al. (2009). Journal of mass spectrometry : JMS, 44(5), 633–640. https://doi.org/10.1002/jms.1536). Indeed, a behavior similar also have been observed with other methods of polymerization (Meneses-Gutiérrez, et al.  (2019). Antioxidants (Basel, Switzerland), 8(7), 214. https://doi.org/10.3390/antiox8070214, Sun, W., & Miller, J. M. (2003). Journal of mass spectrometry : JMS, 38(4), 438–446. https://doi.org/10.1002/jms.456.

Although these procyanidins could be minor components, we would like to hallmark, that their presence in this condition it could be determining the specificity of the anticancer activity that has this procyanidins mix (figure 3D), an important point in the development of new drugs for the cancer treatment.  

the m/z of Figure 2C does not show the existence of an oligomers

R= At this point, we would like to mention that the observation of oligomers with different molecular weight (figure 2C) it can be explaining by the oxidative cascade hypothesis (Kuhnert et al., Mass Rapid Commun. Mass Spectrom. 2010, 24, 3387-3404.

The hypothesis consists of three types of oxidation reactions for the generation of different oligomers: 1) Oligomerization basic type, that is an interflavanic configuration (cross-link between two subunits and structural changes occurring within a molecule as a result of the coupling reaction) that result from one oxidation.

2) Intramolecular rearrangement and tripartite-oligomerizations that always require a second oxidation step. Here, one of the preformed basic interflavanic participates by rearrangement (no elongation; dimeric product) o by forming a scaffold onto which a third catechin is superimposed (An extra oxidation step can also elongate oligomers), simply by attaching subunits

through one of the basic oligomerization types. (Li, Y et al., J. Nat. Prod. 2010, 73, 33-39).

3) Hydroxylation, an extra oxidation step can also elongate oligomers, simply by attaching subunits through one of the basic oligomerization types.

The numerous different combinations of oxidation reactions are thought to cause the extensive molecular diversity of oligomers. An example of this, it is the molecular diversity generated after of the oxidation of catechin in the black tea, where it is estimated the generation of 10,000 different complex phenolics (Kuhnert et al., Rapid Commun. Mass Spectrom. 2010, 24, 3387-3404. Verloop et al. (2016). Journal of agricultural and food chemistry, 64(30), 6011–6023. https://doi.org/10.1021/acs.jafc.6b01695)

Comments on the Quality of English Language

Multiple grammatical mistakes are observed. Some contradictory sentences exists.

  1. Line 44 "Indeed, this procyanidins is found mainly in oligomeric form (degree of polymerization = 10), predominantly dimers to hexamers."

Degree of polymerization would be the average number and it would mean that predominant compounds have polymerization degree of 10 in cocoa. This sentence also requires a reference.

R= El manuscript was submitted English edition and the refences of polymerization degree of 10 in cocoa was included (reference 11).

Round 2

Reviewer 3 Report

Comments and Suggestions for Authors

The authors' response is not acceptable. 

>Although these procyanidins could be minor components, we would like to hallmark, that their presence in this condition it could be determining the specificity of the anticancer activity that has this procyanidins mix (figure 3D), an important point in the development of new drugs for the cancer treatment.

If procyanidins are minor components, how can the authors determine that the anticancer activity arose from the procyanidin in the oxidized mixture?

>R= At this point, we would like to mention that the observation of oligomers with different molecular weight (figure 2C) it can be explaining by the oxidative cascade hypothesis (Kuhnert et al., Mass Rapid Commun. Mass Spectrom. 2010, 24, 3387-3404.

If the authors want to claim that the products are a mixture of oligomeric epicatechins, they should at least submit a presumed structure corresponding to the m/z of the analysis. And moreover, those compounds are not "procyanidins". 

Thus, the title must be revised, the characterization of the oxidized product must be reconsidered, and the logic that "anticancer activity arose from the procyanidin product" must be revised. The authors can only state that the oxidized mixture of epicatechin has anticancer properties.

Author Response

Dear Academic editor.

Thank you very much for your help,

We responded according to observation. We included the chemical structures (new figure 3) and we modify figures and reference number through text. We discuss about the title and in our point of view, we think that is agreeing with the study performed, in fact, the inclusion of the chemical structures is agreeing with the statement in the title. For this reason, we would like to maintain the title.

We appreciated your comprehension and thanked you very much for your help.    

We notice that the main text of the paper is short. As required on our website, the articles should have a minimum of 4000 words (https://www.mdpi.com/about/article_types). Please, try to extend the content while revising by considering these points: adding fully experimental details, presenting completely all the results, and providing comprehensively the background and overview of the research in the introduction section.

R= In the introduction three paragraph were included (line 36-43,47-50, 87-93). These modifications were supported with the references 1-7, 13-15, 36 (they were included in the references).

A new figure was included in the manuscript (Figure 3).

The figure and reference numbers were corrected through text.

Round 3

Reviewer 3 Report

Comments and Suggestions for Authors

I am quite disappointed by the authors response. Authors must realize that they can not claim that the oxidation product is a mixture of procyanidin and that the procyanidins in the mixture showed anti-cancer activity. They can only claim that the mixture contains procyanidin and that the mixture shows anti-cancer activity.

>We included the chemical structures (new figure 3) and we modify figures and reference number through text.

There is no doubt that the authors identified Procyanidin B2, A2 and the trimer from m/z value. What the authors need to show is the structures of others (peak 9-15 in Figure 2C) if they claim they are procyanidins. Authors did not show the structure, which means they are not procyanidins.

>We discuss about the title and in our point of view, we think that is agreeing with the study performed, in fact, the inclusion of the chemical structures is agreeing with the statement in the title. 

Since the majority of the compounds are unidentified, it cannot be said that a mixture of procyanidins was produced by oxidation.

Therefore I think the title must be revised. Example is as follow.

"The autoxidized mixture of (-)-epicatechin contains procyanidins and shows antiproliferative and apoptotic activity in breast cancer cells"

In addition Line 134-137 must be revised. Example is as follow.

The peak with m/z 865.1975 suggested the generation of trimer of (-)-epicatechin (Figure 2C, peaks 11). Other peaks (Figure 2C, peaks 9,10,12-15) may indicate the presence of (-)-epicatechin-derived products of different molecular weights, but their structures could not be identified.

Lastly, conclusion must be revised. Example is as follow

The autoxidation of (-)-epicatechin under mildly acidic conditions (pH = 6) yielded a complex mixture, which presented an antiproliferative activity selective for breast cancer cells. Characterization of the mixture through HPL/MS/MS identified procyanidin types A and B, as well as the presence of trimers of (-)-epicatechin. The antiproliferative effect of the mixture in breast cancer cells was higher than the monomer of (-)-epicatechin, demonstrating that its oxidation affected the anticancer activity. It is important to mention that the anticancer activity of the mixture was found to be mediated by the induction of apoptosis, and they presented selectivity for cancer cells. his study showed that oxidized (-)-epicatechin produced in vitro may be useful in cancer therapy. However, further experiments are necessary to demonstrate their safety and activity in vivo.

Author Response

I am quite disappointed by the authors response. Authors must realize that they can not claim that the oxidation product is a mixture of procyanidin and that the procyanidins in the mixture showed anti-cancer activity. They can only claim that the mixture contains procyanidin and that the mixture shows anti-cancer activity.

>We included the chemical structures (new figure 3) and we modify figures and reference number through text.

There is no doubt that the authors identified Procyanidin B2, A2 and the trimer from m/z value. What the authors need to show is the structures of others (peak 9-15 in Figure 2C) if they claim they are procyanidins. Authors did not show the structure, which means they are not procyanidins.

>We discuss about the title and in our point of view, we think that is agreeing with the study performed, in fact, the inclusion of the chemical structures is agreeing with the statement in the title.

Since the majority of the compounds are unidentified, it cannot be said that a mixture of procyanidins was produced by oxidation.

Therefore I think the title must be revised. Example is as follow.

"The autoxidized mixture of (-)-epicatechin contains procyanidins and shows antiproliferative and apoptotic activity in breast cancer cells"

R= After of an analysis and discussion of your point of view, we agree with the observation, there are some peaks, and we are not sure if they are or not procyanidins. We take your suggestion, and the title was changed.

In addition, Line 134-137 must be revised. Example is as follow.

The peak with m/z 865.1975 suggested the generation of trimer of (-)-epicatechin (Figure 2C, peaks 11). Other peaks (Figure 2C, peaks 9,10,12-15) may indicate the presence of (-)-epicatechin-derived products of different molecular weights, but their structures could not be identified.

R= The lines 134-137 were revised and substituted by the paragraph suggested.

Lastly, conclusion must be revised. Example is as follow

The autoxidation of (-)-epicatechin under mildly acidic conditions (pH = 6) yielded a complex mixture, which presented an antiproliferative activity selective for breast cancer cells. Characterization of the mixture through HPL/MS/MS identified procyanidin types A and B, as well as the presence of trimers of (-)-epicatechin. The antiproliferative effect of the mixture in breast cancer cells was higher than the monomer of (-)-epicatechin, demonstrating that its oxidation affected the anticancer activity. It is important to mention that the anticancer activity of the mixture was found to be mediated by the induction of apoptosis, and they presented selectivity for cancer cells. his study showed that oxidized (-)-epicatechin produced in vitro may be useful in cancer therapy. However, further experiments are necessary to demonstrate their safety and activity in vivo.

R= The conclusion was revised, and the changes suggested was performed.